# Quantification of metabolic niche occupancy dynamics in a Baltic Sea bacterial community

Jana C Massing,[1,2,3] Ashkaan K. Fahimipour,[4] Carina Bunse,[1,3,5] Jarone Pinhassi,[6] Thilo Gross[1,2,3]

**ABSTRACT** Progress in molecular methods has enabled the monitoring of bacterial populations in time. Nevertheless, understanding community dynamics and its links with ecosystem functioning remains challenging due to the tremendous diversity of microorganisms. Conceptual frameworks that make sense of time series of taxonomically rich bacterial communities, regarding their potential ecological function, are needed. A key concept for organizing ecological functions is the niche, the set of strategies that enable a population to persist and define its impacts on the surroundings. Here we present a framework based on manifold learning to organize genomic information into potentially occupied bacterial metabolic niches over time. Manifold learning tries to uncover low-dimensional data structures in high-dimensional data sets that can be used to describe the data in reduced dimensions. We apply the method to re-construct the dynamics of putatively occupied metabolic niches using a long-term bacterial time series from the Baltic Sea, the Linnaeus Microbial Observatory (LMO). The results reveal a relatively low-dimensional space of occupied metabolic niches comprising groups of taxa with similar functional capabilities. Time patterns of occupied niches were strongly driven by seasonality. Some metabolic niches were dominated by one bacterial taxon, whereas others were occupied by multiple taxa, depending on the season. These results illustrate the power of manifold learning approaches to advance our understanding of the links between community composition and functioning in microbial systems.

**IMPORTANCE** The increase in data availability of bacterial communities highlights the need for conceptual frameworks to advance our understanding of these complex and diverse communities alongside the production of such data. To understand the dynamics of these tremendously diverse communities, we need tools to identify overarching strategies and describe their role and function in the ecosystem in a comprehensive way. Here, we show that a manifold learning approach can coarse grain bacterial communities in terms of their metabolic strategies and that we can thereby quantitatively organize genomic information in terms of potentially occupied niches over time. This approach, therefore, advances our understanding of how fluctuations in bacterial abundances and species composition can relate to ecosystem functions and it can facilitate the analysis, monitoring, and future predictions of the development of microbial communities.

**KEYWORDS** niche, marine, manifold learning, diffusion map, bacterial communities

More than 60 years ago, Hutchinson formulated the paradox of plankton, expressing astonishment at the enormous diversity of organisms in the face of an apparently limited number of resources (1). Today, the estimated diversity of microbial species in the ocean extends this apparent contradiction to marine bacterial communities (2, 3). Over 40,000 marine microbial species have been detected so far (4, 5), and these microorganisms are critical for life in the oceans and on land because of their capacity to perform sophisticated and diverse chemical reactions, to drive major biogeochemical

Address correspondence to Jana C Massing, jana.massing@hifmb.de.

The authors declare no conflict of interest.

See the funding table on p. 17.

cycles (6), and to exhibit diverse interactions among each other and with macroorganisms (e.g., 7, 8). In addition to their huge diversity overall, marine bacterial communities undergo complex dynamic fluctuations in abundances and species compositions on the daily, monthly, and annual scale (9, 10). In surface waters in temperate and polar regions, bacterial communities show strong seasonal patterns driven by changes in multiple-interacting environmental features (9).

Due to the recent advances in modern molecular methods it has become possible to monitor bacterial community composition over time and establish short- and long-term time series (examples in 11). These revealed for instance stability in average community composition in spite of strong variation on shorter scales (10), environmental selection as an important driver of seasonal community succession (12), and the significance of biological interactions among bacteria themselves and between bacteria and other organisms, for example, phytoplankton, in shorter-scale community dynamics (13).

Still, the tremendous diversity of bacteria poses a challenge for data analysis. Suppose, for example, that we sample a bacterial community and record the relative population densities for each detected taxon, e.g., amplicon sequence variants, operational taxonomic units, species. In this case, the number of variables per sample is identical to the number of taxa. Mathematically we can say that the dimensionality of the data space equals the number of taxa. Dealing with such high-dimensional data is inherently difficult (the so-called curse of dimensionality) (14). In particular in high-dimensional spaces, comparisons between all but the most similar data points become so noisy that they hurt rather than help the analysis (15). Hence, the analysis can benefit from a coarse-graining step in which the dimensionality of the data set is reduced to a smaller number of variables. If done well, this reduction yields new informative variables and also greatly reduces the noise in the data (16, 17).

When dealing with high-dimensional data spaces, only a tiny portion of space is typically inhabited by data points. In terms of bacterial samples, even though there are many different taxa, we will not see all possible combinations of taxa in the communities. Instead, bacteria appearing together may be characterized by using similar resources (e.g., phytoplankton exudates), profit from certain interactions, or benefit from the same environmental conditions.

Due to these dependencies, the data points may approximately trace a curve, a curved surface, or some other comparatively low-dimensional object within the data space. Such objects in the data space are called data manifolds, and the task of locating them is known as manifold learning. Assume for instance we have a group of bacterial taxa that are metabolically flexible due to a larger genome size, and as a consequence they can use many different carbon sources, hence these taxa become abundant, for example, in response to a phytoplankton bloom. On the other hand, we have a group of taxa with streamlined genomes that are only able to use very few carbon sources and that become dominant in resource-depleted times. In this case, we could abstract from describing each individual taxa and coarse grain these communities in terms of their strategy of degree of trophic versatility with regard to the carbon sources, which would reduce the number of variables used to describe our data. Because the dimensionality of the manifold is lower than that of the embedding data space, manifold learning allows us to reduce the complexity of the data without losing information (18).

A widely used *de facto* manifold learning method is principal component analysis (PCA) (19). PCA is a linear method that approximates the curved manifolds in the data by a flat surface. These surfaces are constructed by a fitting procedure that is implemented as an algebraic operation. Although elegant, this procedure requires long-distance comparisons between data points. The PCA must thus eventually fail in large data sets not only because the manifolds are typically curved but also because long-distance comparisons between data points themselves become unreliable in big data (20, 21).

For illustration, consider the comparison between two species based on the similarity of their genes. If the two species A and B have a 99.9% similarity between their gene sets, then the missing 0.1% offers a good quantification of the difference and we can say,

for example, that B is closer to A than it is to another species C with which it has 0.2% mismatch. On the local scale where such short-distance comparisons are made, even the curvature of the data manifolds becomes negligible and we are *de facto* operating in a flat data space. Contrast this with two species C and D that only have an overlap of 50%. Clearly, we are now comparing very dissimilar species. We could still try to use the 50% difference as a measure of dissimilarity. However, the distance across which we are measuring is now so big that we cannot confidently claim that D is closer to C than it is to another species E to which it is 55% different. The species are so different in this case that such comparisons become virtually meaningless. The underlying mathematical reason is that on this large scale, the curvature of the data manifold needs to be taken into account.

When we naively compare dissimilar genomes, we typically measure the distance between them on the shortest line. Real-world data, however, have non-linear characteristics that cannot be captured by a line. Hence, these long-distance comparisons using the shortest line leave the manifold on which the real genomes lie, and thus leave the space of the biologically plausible. As a result, these long-distance comparisons cannot be trusted unless an effort is made to make such comparisons along the actual data manifold. The diffusion map (22, 23) is a method that identifies the manifolds in a data set and embeds the data in a new space where long-distance measurements faithfully represent dissimilarity measured in the biologically plausible space. To do this, the diffusion map rejects all long-distance comparisons between data points and instead constructs a network where data points are linked to their $k$ most similar neighbors. Once such a network that traces the data manifold has been constructed, the dissimilarity between data points can be computed by an algebraic procedure that is of a complexity comparable to PCA.

A by-product of the diffusion map procedure is a set of so-called eigenvectors. Each of these vectors contains one number for every data point in the data set. We can interpret these numbers as coordinates in a new data space, where faithful long-distance comparisons are possible.

Recent papers demonstrate that the application of diffusion maps to ecological data (20, 21, 24) yields new variables that can be interpreted as composite functional strategies. The diffusion map thus relates to the fundamental ecological niche space in a system (20, 25).

Here we apply diffusion maps to re-construct the metabolic niche space of a bacterial community from a long-term time series in the Baltic Sea. We use the newly predicted variables to convert the taxonomic time series into multiple strategy time series and to quantify changes in functional diversity. Our results indicate that the diffusion map can reveal interpretable ecological strategies in the Baltic Sea bacterial community. This provides a quantitative framework to organize genomic information into potentially occupied ecological niches over time.

## MATERIALS AND METHODS

### Sampling data

Seawater samples were obtained approximately weekly during 2011–2013 and monthly during 2014–2019 from the Linnaeus Microbial Observatory (LMO) (N 56°55.854′, E 17°3.6420′) situated in the Western Baltic Proper. Using 3- or 5-L Ruttner water samplers, water was sampled from 2-m depth at ∼ 9 a.m. during each sampling occasion. Seawater was processed in the laboratory (Linnaeus University) where environmental parameters (temperature, salinity, chlorophyll a, dissolved organic carbon [DOC], nitrate and nitrite [together named as nitrate] , phosphate, silicate, and ammonium) were analyzed as previously described (26, 27, E. Fridolfsson et al., unpublished data). For microbial community composition, we filtered the seawater directly onto 0.22 µm Sterivex cartridge filters (Millipore) or prefiltered onto 3 µm polycarbonate filters and subsequently on 0.22 µm Sterivex cartridges (named 3-0.2 µm size fraction) using a

persistaltic pump. We stored the filters in TE (Tris–EDTA) buffer at −80°C until DNA extraction using a phenol–chloroform method described by Boström et al. (28) and modified after Bunse et al. (29). We amplified the V3V4 region of the 16S rRNA gene using PCRs with the primer pair 341f-805r (30, 31). We analyzed the DNA concentrations with a NanoDrop or Qubit 2.0 Fluorometer (Life Technologies) and ran gel electrophoresis to confirm the amplicon specificity. Sample batches for sequencing were successively sent to the Science for Life Laboratory, Sweden, on the Illumina MiSeq platform, resulting in 2 × 300 bp paired-end reads. For bioinformatic processing, we used the nf-core Ampliseq pipeline (32, 33) with the following software versions: nf-core/ampliseq = v1.2.0dev; Nextflow = v20.10.0; FastQC = v0.11.8; MultiQC = v1.9; Cutadapt = v2.8; and QIIME2 = v2019.10.0. We used DADA2 (34) implemented in QIIME2 (35) and trimmed the sequences at forward 259 bp and reverse 199 bp before denoising. Of all LMO samples, we used all filter fractions for the niche space analysis but only the non-prefiltered 0.22 µm fraction for abundance estimates (see method description given below).

## Obtaining genomes and genes from amplicon sequence variant (ASV) data

We acquired the GTDB reference database (GTDB, https://data.gtdb.ecogenomic.org/, release 95 database) of all 16S rRNA gene sequences from genomes that passed quality control. Sequences were identified with nhmmer (36) using the 16S rRNA model from the RFAM database (37). We then conducted a basic local alignment search tool (BLAST) (38) sequence similarity search using default parameters to match denoised sequence variants present in each LMO sample to the BLAST database and retain the top hits. In the event of multiple matches with equal similarity, we retained a randomly chosen representative. 21,102 (so 44%) of the 48,098 ASVs could be matched well (similarity greater than 95%) with a genome from the GTDB. In terms of abundances, a mean of 82% (column sums) were good matches. From the well-matched species we obtained the complete genomes from GTDB and NCBI (RefSeq and GenBank). These 4,265 complete genomes were annotated using Prokka (39).

## Diffusion mapping of the strategy space

We performed the diffusion mapping following the algorithm described by Barter and Gross (16). This method briefly consists of six steps:

1. Standardize the data.
2. Compute distances between all data points.
3. Construct a similarity matrix.
4. Threshold the similarity matrix.
5. Define a Laplacian matrix.
6. Eigendecomposition of the Laplacian provides new variables.

In our case, the starting point of our analysis is the data table with genomes as rows and annotated genes as columns. If a genome contains a certain gene, the table entry is 1 for the respective row (genome) and column (gene), and if the genome does not contain the gene, the table entry is 0. From this data table, we obtain our data matrix **A** with the dimensions $M \times N$, where $M = 4,265$ is the number of genomes, and $N = 15,361$ is the number of annotated genes. There is no need for standardization of this data set since each variable is considered on the same scale. In the case of heterogeneous variables (e.g., 16), we standardize the data such that each column has a mean of zero and a SD of 1. Second, we compute the hamming distances between all genome pairs counting the differences in gene composition. Thereby, we obtain a $M \times M$ distance matrix **D** containing the hamming distances for each genome pair. The hamming distance is used here due to the presence–absence type of data; if dealing with count data, the Spearman correlation coefficient can be directly used as measure of similarity (21). Third, we define similarities of two genomes as the inverse of the hamming distance of the respective genome pair. The similarity $C_{ij}$ between the two genomes $i$, $j$ is

$$C_{ij} = \frac{1}{D_{i,j}} \tag{1}$$

Thus, we convert our $M \times M$ distance matrix into a $M \times M$ similarity matrix. The diagonal elements comparing each genome with itself are set to zero. This matrix tells us how similar each genome pair is in terms of predicted gene composition. If we imagine the genomes as nodes in a network, we can interpret the similarity matrix as the weight matrix of the network. Fourth, we threshold the similarity matrix, keeping only the top 25 highest similarity entries for each genome and setting all other entries in the matrix to zero. Hence, an entry $C_{ij}$ is kept if it is among the top 25 highest similarity scores for either genome $i$ or genome $j$ or both. Depending on the data set, the threshold has to be adjusted. This is because the aim is to keep the threshold as low as possible without breaking the network (breaking the network results in more than one zero eigenvalue in the final step). Often a threshold between 10 and 30 is optimal, and changes within this range that do not result in breaking the network generally do not impact the results. Fifth, from the thresholded similarity matrix, we compute the corresponding $M \times M$ row-normalized Laplacian matrix, defined by

$$L_{ij} = - \begin{cases} 1 & \text{for } i = j, \\ -\frac{C_{ij}}{\sum_n C_{nj}}, & \text{otherwise}. \end{cases} \tag{2}$$

In the final step, the eigenvectors and corresponding eigenvalues of this Laplacian matrix are computed. There is always one zero eigenvalue and the corresponding eigenvector can be ignored as it contains no information. If the eigendecomposition results in more than one zero eigenvalue, the network has become disconnected and the analysis has to be repeated, setting a higher number of similarity scores to keep in the thresholding step. The eigenvectors define new variables that are composites of metabolic capabilities of the analyzed taxa. The corresponding eigenvalues provide information on the importance of these new variables in describing the structure of our data set. The eigenvector corresponding to the smallest non-zero eigenvalue is the most important, hereafter called variable 1, and the eigenvector corresponding to the second-smallest non-zero eigenvalue is the second most important variable, hereafter called variable 2, and so on. The diffusion map assigns each genome a score for each new variable, for example, the score for genome $i$ in variable 1 is the $i$th entry of eigenvector 1. Since every genome is assigned a score for each variable, we can order the genomes according to their entry, from the most negative to the most positive entries. We interpret these new variables as composite metabolic strategies by examining the genomes that score extreme values in the variable and by checking the enriched genes of the genomes scoring extreme values.

The diffusion map can be applied to almost any data set where some notion of similarity between data points exists. In practice, the data used is often a table that reports a certain number of properties for a certain number of samples. The diffusion map is very robust regarding the size and shape of the data table considered. It is particularly attractive in the case where the number of properties is very high and potentially higher than the number of samples. In this case, the diffusion map is a particularly attractive tool as many alternative methods fail in this case. The diffusion mapping approach remains valid even for very small data sets; however, the insight gained becomes small if the diameter of the network created by the diffusion map after thresholding remains low. Hence we do not recommend the use of diffusion maps for data sets containing less than ca. 30 samples. Conversely, the diffusion map is efficient enough to be applied to very large data sets, with potentially millions of samples. The rate-limiting step is the pairwise comparison of samples, which scales as $O(N^2)$. For extremely large data sets, the complexity may be reduced through the use of heuristics that reduce the number of comparisons that need to be made.

## Translating ASV time series into strategy time series

For analysis, we separated the negative and positive values for each diffusion variable. Taking the relative abundances of the ASVs into account, we calculated abundance-weighted mean values for each variable side. For variable side $v$ and sample $k$, the abundance-weighted mean strategy value is defined as

$$\text{mean strategy value}_{v,k} = \frac{\sum_{i=1}^{n} v_i p_{k,i}}{\sum_{i=1}^{n} p_{k,i}}, \tag{3}$$

where $v_i$ denotes the variable entry of variable side $v$ assigned to genome $i$, $p_{k,i}$ denotes the relative abundance of the ASVs corresponding to genome $i$ in this sample, and $n$ is the number of genomes. To convert the species time series into a strategy time series, we calculated the weighted means of each diffusion variable side for each sample, that is, sampling time-point.

## Identifying over-represented genes

To identify the genes that were over-represented in the genomes of the taxa that themselves were assigned extreme entries along diffusion map variables, we used a permutational variant of the gene set enrichment analysis (GSEA) (40). Genomes were ranked by the orderings specified by each diffusion variable. GSEA was performed using the fgsea library in R (41, 42) with a Benjamini–Hochberg-adjusted (43) P-value <0.01 used as a threshold for retaining genes corresponding to taxa that scored extreme values in the respective diffusion variable.

## Estimating functional diversity

Functional diversity was estimated using the procedure described by Ryabov et al. (21). Briefly, the method uses the Euclidean distances between species in the strategy space, rescaling the eigenvector elements according to their respective eigenvalue to quantify pairwise functional dissimilarities between all species. These newly defined diffusion distances can then be used to obtain the functional diversity of each sample, calculated as the Rao index.

## Data processing

Data were processed with R 4.1.2 (41) and Julia 1.7.1 (44). The two-dimensional mapping of the diffusion variables was made using the package phateR (15). Also the R packages ggplot2 (45), tidyverse (46), plyr (47), ggbreak (48), ggpubr (49), seriation (50), and reshape (51) were used.

## RESULTS AND DISCUSSION

Our goal is to discover changes in the functional potential of marine bacterial communities in time. The ideal data for this task would be time series of sequenced isolate genomes for the same marine community, censused repeatedly. However, these data are generally not available. We, therefore, illustrate the method using a straightforward mapping between 16S rRNA gene amplicon sequencing time series and closely related and publicly available microbial genomes.

Our primary analysis does not use external data on gene function. Instead, we use the genes to define a primary notion of similarity between species. We compute the dissimilarity of a pair as the hamming distance in known gene composition. This measure of dissimilarity is only trustworthy for comparisons between similar species, but only such short-ranged comparisons are needed for a diffusion map analysis. Running the data through the diffusion map identifies new functional coordinate axes i.e., new variables. The diffusion map ranks these variables from the most to the least important one. Hence, in the following, we refer to variable 1 as the most important, variable 2 the second most important, and so on.

The new variables are composites of metabolic capabilities of the analyzed taxa i.e., metabolic strategies. Each axis represents a dimension of the metabolic strategy space (20). These new variables can trace a continuum in the strategy space ranging from one pure strategy at one end to another pure strategy at the other. The diffusion map gives us such variables if there is a true continuum in the data. For example, starting with genomes that have the full set of genes for a given pathway at one end of the dimension and then going to partial sets and eventually to another pathway for which the set of genes is gradually completed as we near the other end of the dimension. Thus, we get such variables specifically if there is no natural cutoff. In fact, that such results are possible is a clear advantage of manifold learning over clustering approaches, which need to introduce a cutoff artificially. We divide each variable into positive and negative side, where the terms "positive" and "negative" are used to imply that there is a continuous gradual spectrum between two strategies.

If a natural separation exists in the underlying data (e.g., a pathway is almost complete or almost completely absent, with little middle ground), then the diffusion map will return localized eigenvectors. In this case, there is a clear separation between non-zero (feature present) and almost zero (feature absent) entries of the vector, such that the cutoff point is easy to identify.

As each taxon from the data set is assigned a score for each new variable, we can use these scores to order taxa from the most negative to the most positive entries for each variable. By analyzing the functional capabilities of taxa near the axis ends, we can interpret the metabolic strategies that are described by the newly identified variables, as these taxa represent the most clear examples of the corresponding function. In finite noisy data, there can be some cases where the distinction is not as clearcut as we described it here, but it is generally easy to find a good operational definition of what is considered to be near which end, and the results prove to be insensitive to how exactly this choice is made.

The eigenvectors provide us with scores for each taxon in each strategy that we can use to convert our taxonomic time series into multiple strategy time series. These are computed by calculating the abundance-weighted means of each variable side for each time point (Fig. 1). This enables us to observe the dynamics of community composition over time in terms of the dynamics of putatively occupied metabolic strategies, that is, niches. Altogether, the newly identified variables span the metabolic niche space of the analyzed community.

## Important metabolic strategies in the Baltic Sea data

The most important variable identified via diffusion mapping, variable 1, separates primarily animal- or human-associated members of the Enterobacteriaceae, for example, close relatives of *Salmonella*, *Cronobacter*, and *Yersinia* from all other taxa (Table S2). A collection of 92 genomes from representatives of the Enterobacteriaceae scored low values, whereas all other genomes are assigned values near zero (Fig. 2A). Such a variable

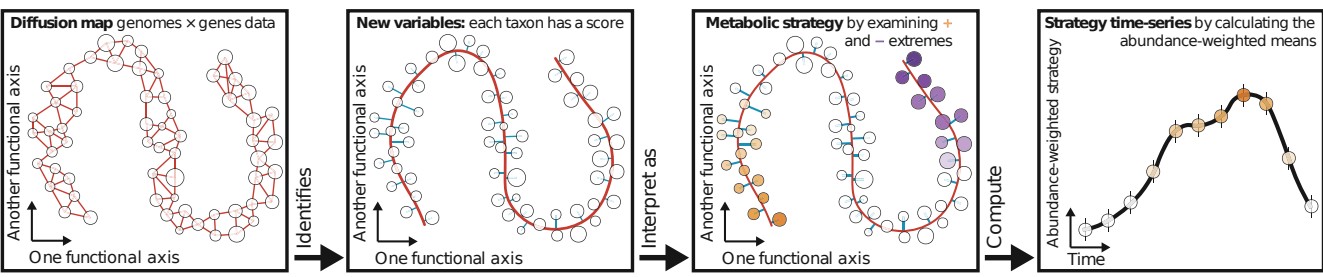

FIG 1 Overview of the procedure from diffusion mapping the data set of genomes and genes to conversion of the species time series into strategy time series. The circles represent the genomes, which have different abundances over time (size). The two axes in the figure are just representatives for the many axes that describe the known gene content of the genomes in a high-dimensional space.

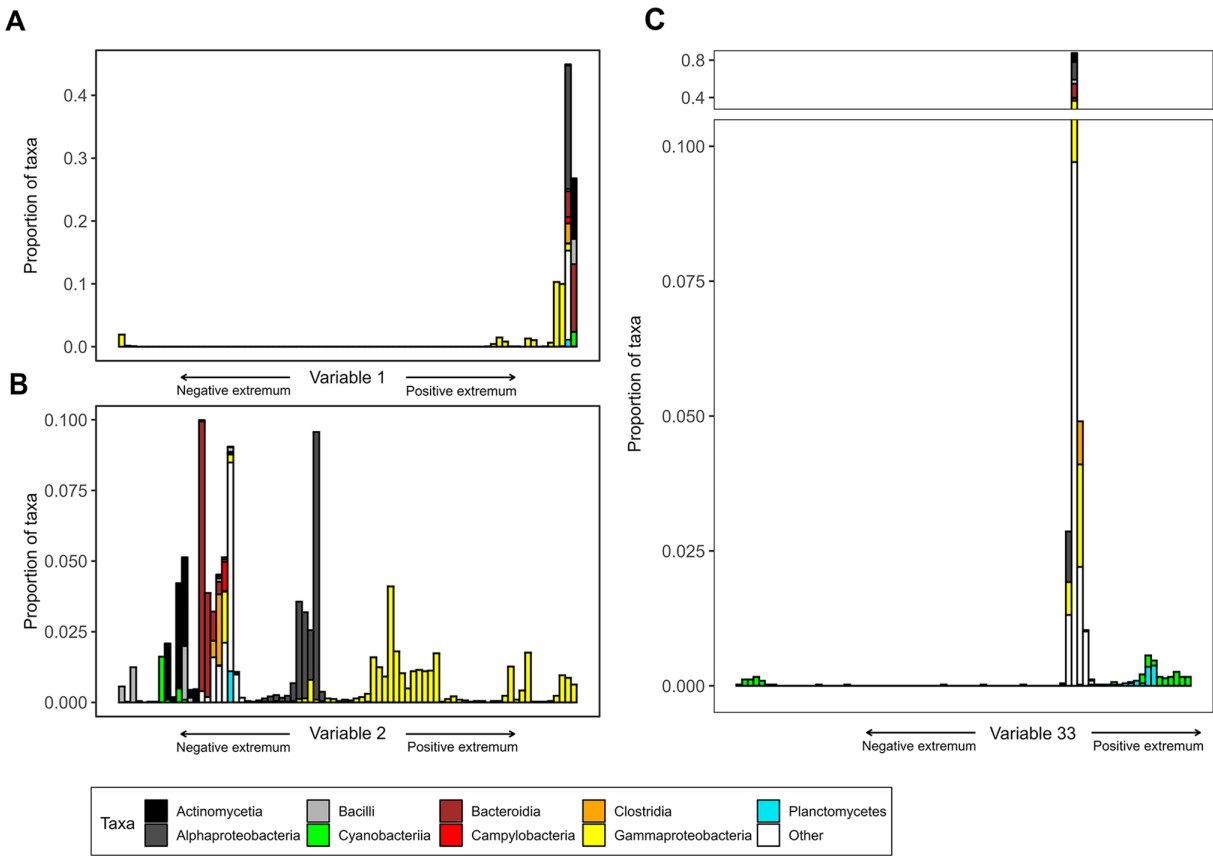

**FIG 2** The ordering of taxa defined by variable 1 (A), variable 2 (B), and variable 33 (C) entries, from negative to positive (left to right). The taxonomic compositions corresponding to variable entries are shown for each of 80 equally spaced bins.

that identifies a clear cluster of species that is well separated from the rest is called a "localized" variable of the diffusion map (52). Localized variables indicate discrete clusters of unique capabilities of the respective taxa.

To reveal which metabolic capabilities characterize these taxa and distinguish them from all others, we identified genes that were over-represented in the genomes of the taxa scoring far-from-zero entries (see Materials and Methods). Enriched genes in these taxa are genes encoding machinery for iron acquisition common in Enterobacteriaceae, such as the Enterobactin synthase component *F* (53, 54), genes responsible for the flagellar formation (55), and genes associated with biofilm formation (56) (Table S3A).

Despite very low abundances of the respective taxa in the samples (mean relative abundances of 0.007 over all samples; see also Fig. S1 and Table S1 for abundance data), the first variable appears first in the diffusion map. This may at first appear as a surprising result given what is known about the ecological role of Enterobacteriaceae in marine ecosystems. However, it makes sense that this group is so clearly separated from the other bacterial taxa due to well-known biases toward sequences of pathogenic taxa and genes involved in pathogenesis in global databases (e.g., 57). Consequently, these taxa are characterized as very different to other bacterial species in the marine community. Because the "full" size fraction was sampled, we might have captured plankton pathogens (e.g., from zooplankton microbiomes or similar) that are very close relatives to these Enterobacteriaceae separated by the first diffusion variable. Hence, this variable separates the most different bacterial taxa in terms of their known gene composition from the rest of the community, demonstrating the power of the diffusion map method to reveal such differences and to identify biases in the data set. The ability to reveal such biases in the data and separate them from the salient information is a strength of the diffusion map.

The diffusion map reveals further localized variables that represent relevant metabolic strategies for the Baltic Sea bacterial community, for example, variable 4 negative, which separates the Cyanobacteria from all other taxa (Fig. S2B). The cyanobacterial genomes are assigned large negative values, whereas all other taxa score positive or close-to-zero values. Genes that encode the subunits of photosystem I and photosystem II as well as associated cytochrome components and cyanobacterial-specific light-harvesting antennae (58) are among the enriched genes, indicating that this variable detects cyanobacterial photosynthesis (Table S3D). Supporting the findings of a previous study (20), the localized character of this variable reveals that cyanobacterial photosynthesis is a yes-or-no strategy, indicating that this photosynthetic lifestyle has wide-ranging metabolic consequences with distinct implications. For example, an oxygenic photosynthetic lifestyle is expensive in terms of avoiding or repairing photoinhibition and damage. Therefore, many costly adaptations are necessary and the energy spent cannot be invested into other metabolic pathways (59).

There are also variables that span a continuum of strategies, such as variables 2 and 3. In variable 2, marine host-associated Gammaproteobacteria, for example, *Vibrio*, *Shewanella,* and *Photobacterium*, are found at the positive extremum, whereas oligotrophic Gammaproteobacteria and Alphaproteobacteria are assigned values close to zero (Fig. 2B). Among the most correlated capabilities for the taxa at the positive end of variable 2 are chemotaxis and response to various stressors (Table S3B). Variable 3 identifies the different strategies of marine Alphaproteobacteria: we find the Rhodobacteraceae and the Rhizobiales, known for their capability of utilizing a variety of carbon sources (60, 61), at the positive extremum (Fig. S2A). Major enriched genes encode machinery for the utilization of various dissolved organic compounds (Table S3C), for example, phosphonate, acetate, and urea that constitute exudates of phytoplankton (62). The streamlined genomes of the free-living Pelagibacterales and the obligate intracellular pathogens Rickettsiales score close-to-zero values. Hence, we interpret variable 2 positive as the metabolic strategy of marine host-associated Gammaproteobacteria and variable 3 positive as the metabolic strategy of Alphaproteobacteria capable of using a wide range of carbon sources.

Taxa that are grouped together in one variable can be separated by another variable. For example, Cyanobacteria group together in variable 4 negative, whereas they are split in variable 33: the Picocyanobacteria score extreme negative values, whereas the other cyanobacterial genomes group toward the positive side, with the heterocyst-forming family Nostocaceae scoring highest values (Fig. 2C and Fig. S2E). The Enterobacterales that score highest in variable 2 are separated in variable 27, for which the family Shewanellaceae scores extreme negative values and the other enterobacterial families, for example, the Vibrionaceae score positive values (Fig. S2D). Variable 14 separates the Bacteroidota into anaerobic, intestinal Bacteroidota, for example, *Prevotella* (63) on the positive side and complex polysaccharide degraders, for example, *Flavobacterium* (64) on the negative side (Fig. S2C). Enriched genes for the latter encode different CAZymes (carbohydrate-active enzymes) (65), responsible for the degradation of major plant cell wall components (Table S3E).

Diffusion mapping also identified strategies that group genera from different taxonomic groups together, we find, for example, bacteria that oxidize methyl groups and C1 compounds, such as methanol and formaldehyde from different families such as Beijerinckiaceae, Xanthobacteraceae, and Acetobacteraceae at the negative end of variable 38 (Fig. S2F). Most correlated genes (Table S3F) encode machinery for methanol and formaldehyde degradation (66). Another example is variable 43 positive that groups the non-spore-forming sulfate-reducing bacteria from the families Desulfocapsaceae, Desulfobacteraceae, Desulfovibrionaceae, and others together (Fig. S2G). Enriched genes (Table S3G) are responsible for sulfate respiration (67).

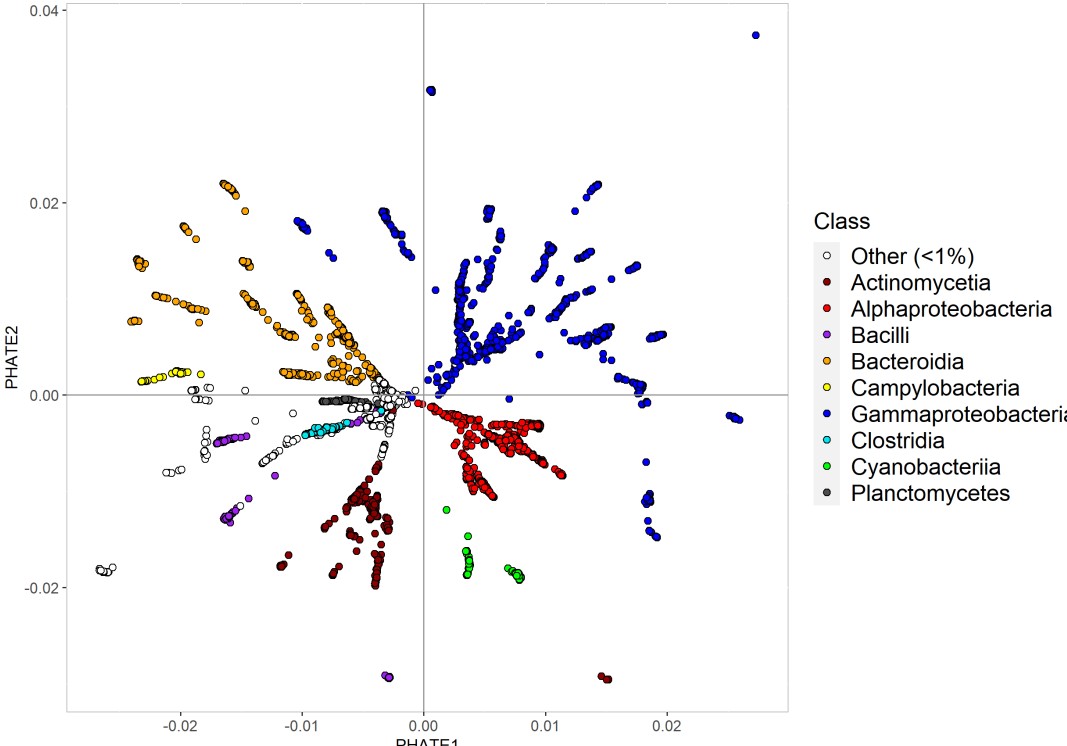

**FIG 3** Two-dimensional embedding of diffusion variables created using the PHATE algorithm (15). Each point represents an individual genome that is colored by taxonomic class.

## Structure of the inferred niche space

The examples from above demonstrate that the diffusion map finds reasonable metabolic strategies of the bacterial community. Jointly, these strategies define the metabolic niche space of the community and assign the taxa to specific coordinates in a multidimensional space.

The diffusion map unfolds the data manifold in a still relatively high-dimensional space. Particularly, if the main manifold is a tree-like branched structure, each of the branches will point in a different dimension. For quantitative analysis, this is very useful because this relatively high-dimensional representation means that metric distances in this representation accurately quantify functional difference (e.g., 21). However, if we want a more intuitive understanding of the structure of the data set, the diffusion map alone is not so helpful as, say, the 20 major branches of a tree would be pointing in 20 different dimensions. However, the tree structure can be represented in two dimensions. This is the idea of the PHATE (Potential of Heat-diffusion for Affinity-based Transition Embedding) (15): multidimensional scaling is used to find a good two- or three-dimensional embedding for the diffusion map results. PHATE, therefore, is a useful tool to visualize high-dimensional data.

Although it is important to interpret low-dimensional embeddings cautiously, as in previous work, the strategy space of the Baltic Sea bacterial community appears as a tree-like structure comprising clusters of taxa featuring localized strategies and continuous branches (20) (Fig. 3). The geometric structure implies large areas of the metabolic niche space that are unoccupied, reflecting either strategies that have yet to be discovered or strategies that are not feasible in the respective ecosystem or even in general (20).

The structure of the metabolic niche space roughly aligns with the phylogeny of the taxa. Uncultured taxa and the streamlined genomes of Patescibacteria (68), Pelagibacterales (69), and Rickettsiales (70) form the core from which the branched structure

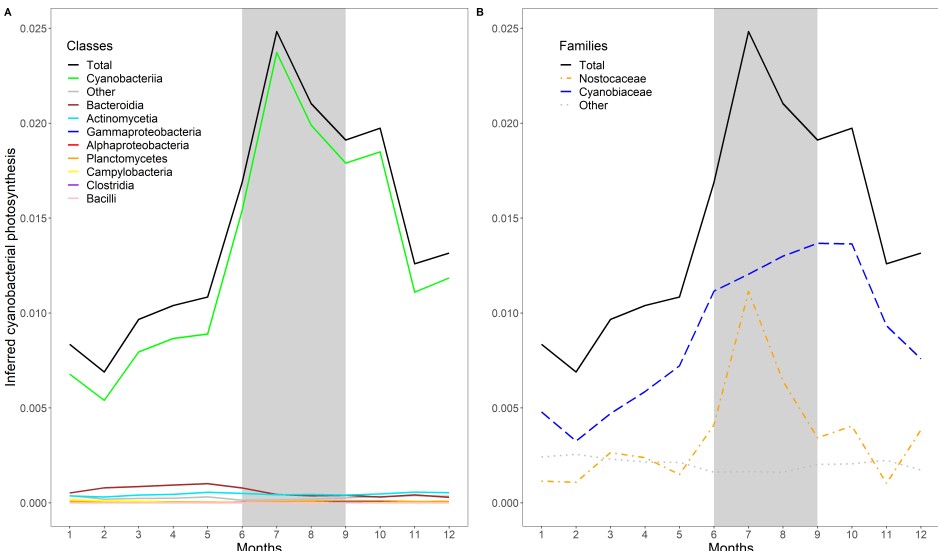

**FIG 4** Strategy time series with different taxonomic resolutions: abundance-weighted mean values of inferred cyanobacterial photosynthesis over the yearly cycle. Summer months are indicated by a gray background. Taxonomic class (A) and cyanobacterial taxonomic families (B) are color-coded.

emerges. The position of the streamlined genomes seems reasonable as these genomes retain mostly the key basic functions necessary for survival and reproduction that they share with many other organisms and they lack many of the more "specialized" genes (71). The uncultured taxa are grouped to the center either because they possess streamlined genomes or due to the lack of knowledge about their genes and respective functions. Therefore, their position in the metabolic niche space might change with further knowledge gained.

Taxa that are characterized by localized variables such as the Cyanobacteria are more distinct within the structure. These distinct clusters could indicate that complex, costly adaptions are necessary to follow the respective metabolic strategy and the machinery cannot easily be acquired, for example, through horizontal gene transfer (72). Localized strategies could also reflect that an intermediate strategy is not feasible, for instance due to certain trade-offs resulting from adopting the respective strategy (73).

Bacterial taxa associated with human disease, like relatives of *Klebsiella*, *Mycobacterium*, *Staphylococcus,* and *Fusobacterium*, group furthest away from the main structure and appear as clusters of dots in the periphery (Fig. 3). The reason for these taxa to group away is probably the bias in global databases toward human pathogens and their functional capabilities (see above).

## Metabolic strategies over time in the Baltic Sea

Above, we saw that the new variables identified by diffusion mapping represent interpretable metabolic strategies of the sampled community. Taking the relative abundances of the taxa that map to different genomes into account, we calculated the abundance-weighted means of each diffusion variable side for each sampling time point (see Materials and Methods). This enables us to observe how the occupation of metabolic niches or strategies changes over time in the Baltic Sea bacterial community.

For example, the variable 4 negative, identified as cyanobacterial photosynthesis, reaches its highest abundance-weighted niche values in summer (Fig. 4). Cyanobacteria are known to cause massive summer blooms in the Baltic Sea (74), supporting our interpretation that this strategy represents cyanobacterial photosynthesis. Separating the different cyanobacterial families reveals an early summer peak caused by the filamentous Nostocaceae and a plateau high of the niche values for the unicellular Cyanobiaceae increasing until the beginning of autumn (Fig. 4B). Utilization of nutrients

from filamentous Cyanobacteria might fuel the metabolism of opportunistic picocyano-bacteria (75).

Another strategy that is affected by seasonality is the catabolism of complex polysaccharides, identified in the variable 14 negative. Dominated by members of the Flavobacteriaceae, this strategy reaches the highest abundance-weighted mean values in May (Fig. 5B and Fig. S4), following the peak of the phytoplankton spring bloom (26). Large amounts of photosynthetic products, mainly polysaccharides, are exuded by phytoplankton (76) and Flavobacteria are adapted to use these high-molecular-weight molecules (64). In spring 2011 coinciding with the highest phytoplankton biomass (26), this strategy reaches high values (Fig. 5A).

Variable 3 positive, describing the ability to utilize a variety of carbon sources, also reaches its highest values in May but does not show a pronounced peak and instead decreases more slowly, reaching a minimum mean value in September (Fig. 5D and Fig. S3). This strategy is dominated by the marine Rhodobacteraceae that are crucial in processing low-molecular-weight phytoplankton-derived metabolites and characterized by their high trophic versatility (77, 78). In winter 2015/2016 and 2016/2017, elevated values of this variable 3 positive are driven by Rhodospirillales, especially of the genus *Thalassospira* (Fig. 5C). The latter is known for its ability to degrade polycyclic aromatic hydrocarbons (79), and their appearance in the upper water column could be related to Major Baltic Inflow events (80) and subsequent winter mixing.

In contrast to the strategies we discussed above that were positively impacted by the seasonal phytoplankton spring bloom, variable 38 negative reaches its minimum in May, right after the phytoplankton bloom (Fig. S5). Describing the metabolic ability to oxidize methyl groups and C1 compounds, such as methanol and formaldehyde, this strategy is dominated by Alphaproteobacteria, especially *Pelagibacter* in winter and Planctomycetes in autumn in the Baltic Sea bacterial community. Marine DOC is a source for diverse C1 and methylated compounds; methanol constitutes a major fraction of oxygenated volatile organic chemicals; and formaldehyde is omnipresent in seawater (66). The ability to use these compounds enables energy production from relatively abundant substrates in the water, but this ability is outcompeted when concentrations of phytoplankton-derived substrates increase in spring. Variable 43 positive describing non-spore-forming sulfate reducers also reaches its minimum mean value in May (Fig. S6). Baltic Sea sulfate reducers of the phylum Desulfobacterota, inhabiting mostly sediments and oxygen-

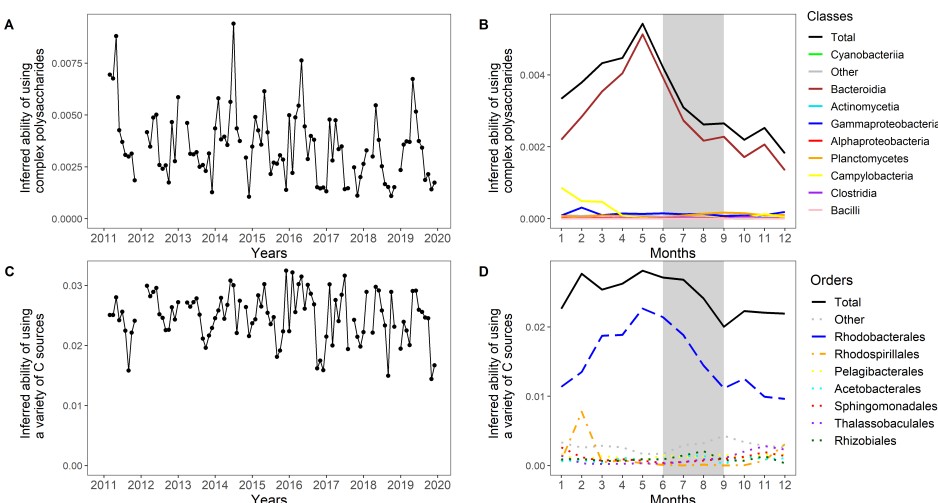

**FIG 5** Comparing two strategies over time: abundance-weighted mean values of inferred ability of using complex polysaccharides (A, B) and inferred ability of using a variety of carbon sources (C, D) over the whole time period and over the yearly cycle. Summer months are indicated by a gray background. Taxonomic orders (B) and taxonomic classes (D) are color-coded.

depleted water (81, 82), drive the peak of this strategy in February, probably appearing in the upper water column due to strong winter mixing.

Comparing the strategy time series to the environmental data obtained at LMO and the abundance-weighted mean values over the months, we can see a strong signal of seasonality (Fig. 6): the variables on the left side of Fig. 6A are correlated with rather high nutrient concentrations and lower temperatures, hence winter conditions, whereas the variables on the right side of the heatmap are correlated with higher temperatures and chlorophyll a concentrations, hence summer conditions. Fig. 6B complements this picture: on the left side, we find higher variable values in summer and autumn, while the variables on the right-hand side show higher values in winter and spring.

## Functional diversity

Due to the global climate and biodiversity crisis, there is a strong need for robust functional diversity monitoring for ecological communities (83). The Rao index (84, 85) computes functional diversity from pairwise functional distances between species.

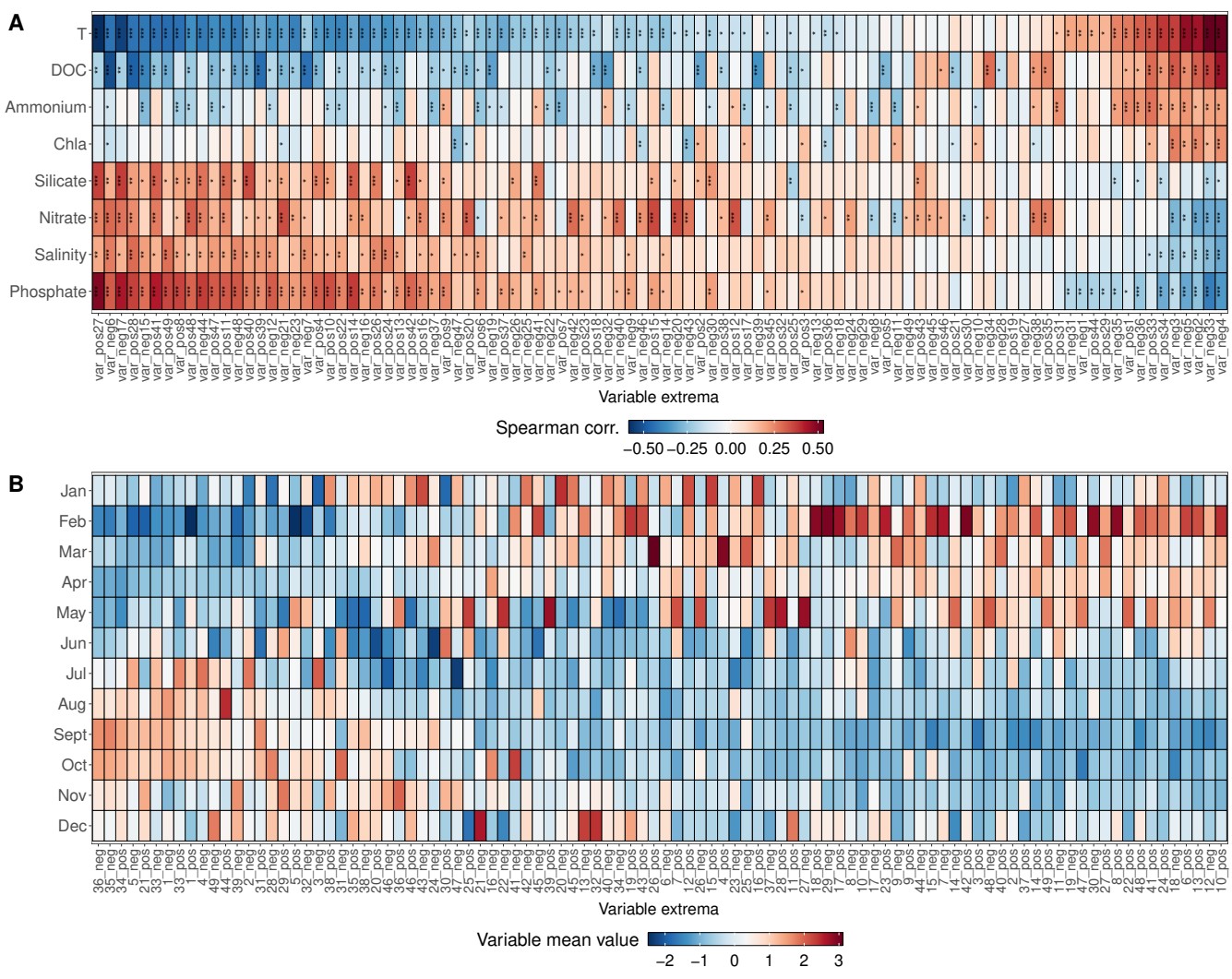

**FIG 6** Signal of seasonality in strategy time series: (A) Heatmap of Spearman correlation coefficients (CV) between the first 49 variables, that is, strategy time series and the environmental variables (metadata). Significance levels expressed by asterisks (***P-value ≤ 0.001, **P-value ≤ 0.01, and *P-value ≤ 0.05). P-values are Benjamini–Hochberg adjusted (43). (B) Heatmap of abundance-weighted variable mean values of the first 49 variables for each month over the whole sampling period, standardized to mean = 0 and SD = 1 for each variable side. Chla: chlorophyll a concentrations; DOC: dissolved organic carbon concentrations; T: temperature. All concentrations are expressed in µM.

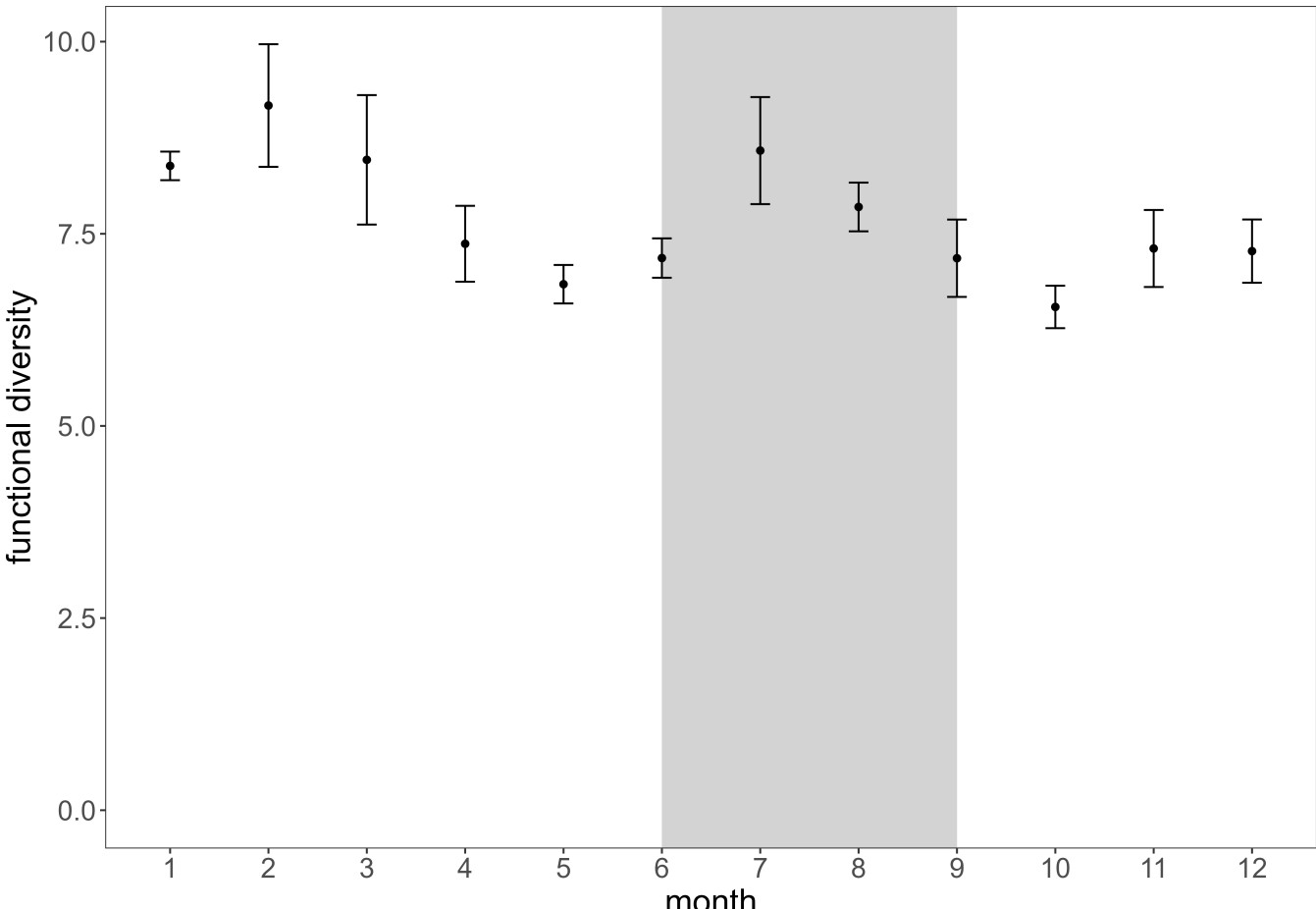

**FIG 7**  Mean and SE of functional diversity estimation calculated as Rao index (21) for the sampled Baltic Sea bacterial community over the yearly cycle summarizing the years 2011–2019. Gray background indicates summer months.

Typically, researchers need to first identify functional traits of the species of interest to be able to define such distances in trait space (e.g., 86). Ryabov et al. (21) have shown that using the diffusion distances in the reconstructed trait space can robustly estimate functional diversities of communities. Following their approach, from the variables obtained via diffusion mapping, we derived the diffusion distances between all pairs of species to quantify the functional diversity of each sample calculated as Rao index (21) (see Materials and Methods). High functional diversity values indicate that the sampled community is diverse in terms of their metabolic strategies, whereas lower values indicate that the community is dominated by fewer metabolic strategies.

The yearly cycle summarized over all sampling years shows the highest functional diversity mean values in February and July, whereas it reaches lowest values in May and October (Fig. 7). This pattern is likely explained by the absence of the thermocline in winter, which causes deeper mixing of the water layers (87), resulting in bacterial communities from the former mesopelagic to be found also in surface waters (27, 88). Members of these communities can possess very different strategies due to their adaptation to deeper water layer or sediment conditions (27), and their taxonomic and functional diversities are in general higher than in surface communities (4), hence they increase the functional diversity when they are observed in the surface layers in winter. Functional diversity also benefits from increased diversity of nutrients available in winter, leading to an increased resource heterogeneity (89, 90). Lower values of functional diversity in May, following the phytoplankton bloom, can be explained by the strong dominance of strategies that relate to the utilization of phytoplankton-derived substrates

during bloom phases as these compounds increase drastically in relative abundance (64, 65, 91).

## Conclusion

In this paper, we showed that the relative abundance of prokaryotic populations obtained from amplicon sequencing data from monitoring data sets can be translated into potentially occupied ecological niches over time. The diffusion map of the bacterial capabilities detected a wide spectrum of interpretable metabolic strategies of the Baltic Sea bacterial community: from localized strategies such as cyanobacterial photosynthesis to continua of strategies such as degree of association with marine hosts and degree of trophic versatility, revealing both, strategies that align with phylogeny, strategies that differentiate closely related taxa, and similar strategies among distantly related taxa (20). The latter may reflect metabolic niche convergence (92) or horizontal gene transfer (e.g., 93).

Systematizing the genomic information via our diffusion map approach provided the possibility to express the changes in bacterial species abundances as quantitative changes in potential occupation of metabolic niches over time. These here called abundance-weighted strategy values showed a variety of patterns in time: strategies displaying seasonal dynamics, for example, increasing trends in summer or an increase following the phytoplankton bloom or higher values as a consequence of winter mixing as well as strategies showing interannual changes in time patterns. Impacts of single events, for example, a very pronounced spring bloom or a Major Baltic Inflow event, were also reflected in the strategy time series. Some functional strategies are clearly dominated by one bacterial group, whereas others are divided between bacterial groups often depending on the season.

Overall, seasonality seems to be a strong driver not only for phylogenetic bacterial composition (27, 88) but also for bacterial metabolic niche occupation and functional diversity of bacterial communities in the Baltic Sea. Seasonality is probably such an important driver also for metabolic strategies due to the interplay of seasonal changes in substrate availabilities together with changes in abiotic parameters that influence metabolic activities (88, 94, 95).

The first new variable identified by the diffusion map separates the metabolic strategies of pathogenic members of the Enterobacteriaceae from all other taxa. The reason for this variable showing up first is that the diffusion map probably detects the bias that bacterial pathogenic taxa and genes involved in pathogenesis are over-represented in global databases (e.g., 57). This makes clear that a major limitation of this approach is our understanding of genes and their functions, implying also that the metabolic niche space can change due to further knowledge gained. For example, we may observe more branches representing currently undetected taxa or strategies and taxa currently grouped toward the center might shift further outwards and distances between taxa might change with increasing information. On the other hand, this result also highlights the power of the diffusion map approach to objectively detect such biases in the data.

It should be noted that genes only encode the theoretical capabilities of a species (20), conceptually corresponding to the fundamental niche concept (25). It was, however, shown that functional genes can be used to predict the position of species along major niche gradients, outperforming the predictions based on phylogenetic information (96). In addition, Gralka et al. (97) demonstrated that functional predictions from genomes very well predict real metabolic niches. Gowda et al. (98) also showed for the process of denitrification that it is possible to predict community metabolic dynamics from the presence and absence of genes in metagenomes. In comparison to their studies, our approach of assigning ASVs to species and obtaining their complete genomes from a database relies heavily on the quality of the databases and genome assemblages available. The similarity of 95% that we used to match ASVs to genomes is a broad identification, and there can be substantial change in genome content at the subgenus

level (e.g., 99). To our knowledge, ASV data at the moment are the only data available for this system that suffices the data needs for our method. As described above, already this crude tool provides evidence that the method yields a useful trait space.

In the future, diffusion map analysis of bacterial trait space may profit from several ongoing developments. The analysis of metagenomes is a desirable tool to improve our analysis by accounting also for within-species genome variability, the accessory genome, and prophages that can play a role in shaping the strategies and traits of the respective organisms (100, 101, 102). Diffusion mapping may also be used in conjunction with tools for the prediction of metagenome functions, for example, PICRUSt 2 (103). Ideally, we would have all the genomes available from exactly the taxa from the habitat sampled, but as we do not, we rely on a simple mapping scheme. Deep shotgun metagenomic sequencing and long-read technologies will be important for collecting the sorts of data that would make our method even more powerful. We also envision that with future transcriptomic data, it could also be possible to investigate the strategies that are deployed under a given set of environmental conditions (20).

We conclude that the diffusion map approach presented here enables us to coarse grain complex bacterial communities in terms of reasonable metabolic strategies and provides a quantitative framework to organize genomic information into potentially occupied metabolic niches over time. Thereby, this approach enables us to understand the dynamics of community composition on different scales in terms of their impact on potentially occupied metabolic niches and to link genomic data to metabolic strategies enhancing our understanding of the relationships between community compositions and ecosystem functions.

## ACKNOWLEDGMENTS

We thank the many people involved in LMO sampling and laboratory work over the years, especially Sabina Arnautovic, Anders Månsson, Emil Fridolfsson, Benjamin Pontiller, and Kristofer Bergström. Furthermore, we acknowledge the help and assistance from Northern Offshore Services (NOS), M/V Provider crew, E.ON and RWE on the samplings.

We acknowledge Daniel Lundin for bioinformatic support.

The LMO research was supported by the Swedish Research Council FORMAS Strong Research environment EcoChange (Ecosystem dynamics in the Baltic Sea in a changing climate) to J.P. C.B., J.C.M., and T.G. were supported by the Helmholtz Institute for Functional Marine Biodiversity (HIFMB), a collaboration between the Alfred-Wegener-Institute, Helmholtz-Center for Polar and Marine Research, and the Carl-von-Ossietzky University Oldenburg, initially funded by the Ministry of Science and Culture of Lower Saxony (HIFMB Project) and the Volkswagen Foundation through the "Niedersächsisches Vorab" grant program (grant no. ZN3285). A.K.F. was supported by the Research Associateship Program from the National Research Council of the National Academies of Sciences, Engineering, and Mathematics; and by National Science Foundation grant EF-2222478. We acknowledge support by the Open Access Publication Funds of Alfred-Wegener-Institut Helmholtz-Zentrum für Polar-und Meeresforschung.

## AUTHOR AFFILIATIONS

[1]Helmholtz Institute for Functional Marine Biodiversity (HIFMB) at the University of Oldenburg, Oldenburg, Germany

[2]Helmholtz Centre for Marine and Polar Research, Alfred-Wegener-Institute, Bremerhaven, Germany

[3]Institute for Chemistry and Biology of the Marine Environment (ICBM) Carl-von-Ossietzky University, Oldenburg, Germany

[4]Department of Biological Sciences, Florida Atlantic University, Boca Raton, Florida, USA

[5]Department of Marine Sciences, University of Gothenburg, Gothenburg, Sweden

[6]Centre for Ecology and Evolution in Microbial Model Systems (EEMiS), Linnaeus University, Kalmar, Sweden

## AUTHOR ORCIDs

Ashkaan K. Fahimipour http://orcid.org/0000-0001-9138-3593
Jarone Pinhassi http://orcid.org/0000-0002-6405-1347

## FUNDING

| Funder | Grant(s) | Author(s) |
| --- | --- | --- |
| Niedersächsisches Ministerium für Wissenschaft und Kultur | HIFMB project | Carina Bunse |
| | | Thilo Gross |
| | | Jana C Massing |
| Volkswagen Foundation | ZN3285 | Carina Bunse |
| | | Thilo Gross |
| | | Jana C Massing |
| Svenska Forskningsrådet Formas | EcoChange | Jarone Pinhassi |
| National Science Foundation | EF-2222478 | Ashkaan K. Fahimipour |

## DATA AVAILABILITY STATEMENT

The LMO 16S rRNA raw data used in this study are deposited in the EMBL-EBI European Nucleotide Archive repository (https://www.ebi.ac.uk/ena) under accession numbers PRJEB52828, PRJEB52855, PRJEB52837, PRJEB42455, PRJEB52851, PRJEB52854, PRJEB52627, PRJEB52772, PRJEB52496, PRJEB52780, PRJEB52782, PRJEB52850, PRJEB56744, PRJEB56745, and PRJNA260662. The environmental data set is available from authors upon request.

## ADDITIONAL FILES

The following material is available online.

### Supplemental Material

**FIG S1 (298436_1_supp_6669226_rr8k6k.tif).** Relative mean abundances of classes that map to the genomes obtained from amplicon sequencing data over the whole sampling period. Taxonomic classes are color-coded.

**FIG S2 (298436_1_supp_6669232_rr8v6p.tif).** The ordering of taxa defined by variable 3 entries (A), variable 4 entries (B), variable 14 entries (C), variable 27 entries (D), variable 33 entries (E), variable 38 entries (F), variable 43 entries (G), from negative to positive (left to right). The taxonomic compositions corresponding to variable entries are shown for each of 80 equally spaced bins.

**FIG S3 (298436_1_supp_6669233_rr846p.tif).** Abundance-weighted mean values of inferred ability of utilizing a variety of carbon sources over the yearly cycle. Summer months are indicated by a gray background. Taxonomic class (A) and taxonomic orders (B) are color-coded.

**FIG S4 (298436_1_supp_6669230_rr8y6p.tif).** Abundance-weighted mean values of inferred ability of degrading complex polysaccharides over the yearly cycle. Summer months are indicated by a gray background. Taxonomic class (A) and taxonomic families (B) are color-coded.

**FIG S5 (298436_1_supp_6669229_rr8l6p.tif).** Abundance-weighted mean values of inferred ability of oxidizing methyl groups and C1 compounds over the yearly cycle. Summer months are indicated by a gray background. Taxonomic class is color-coded.

**FIG S6 (298436_1_supp_6669228_rr8p6p.tif).** Abundance-weighted mean values of trait dominated by non-spore forming sulfate reducers over the yearly cycle. Summer months are indicated by a gray background. Taxonomic class is color-coded.

**TABLE S1 (298436_1_supp_6669231_rr896p.pdf).** Genomes that map to the 100 most abundant ASVs obtained from amplicon sequencing data in terms of relative mean

abundance over the whole sampling period. Genome, taxonomic information, mean and maximum abundance over the whole sampling period are provided.

**TABLE S2 (298436_1_supp_6669249_rr826q.pdf).** Species that map to the 100 ASVs scoring most negative values in variable 1.

**TABLE S3 (298436_1_supp_6669227_rr8x6p.pdf).** Top 100 over-represented annotated genes in the genomes of the taxa that receive the most negative entries on variable 1 negative (A), variable 2 positive (B), variable 3 positive (C), variable 4 negative (D), variable 14 negative (E), variable 38 negative (F), variable 43 positive (G). The NES and FDR-Adj. P columns show the normalized 'Enrichment score' and FDR-adjusted (43) P-value from the enrichment analysis (40).

## Open Peer Review

**PEER REVIEW HISTORY (review-history.pdf).** An accounting of the reviewer feedback and comments.

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
