## [Reviewer comments · mSystems]

Quantification of metabolic niche occupancy dynamics in a Baltic Sea bacterial community

Jana Massing, Ashkaan Fahimipour, Carina Bunse, Jarone Pinhassi, and Thilo Gross

Corresponding Author(s): Jana Massing, Helmholtz-Institut für Funktionelle Marine Biodiversität an der Universität Oldenburg

Review Timeline:

Submission Date:	January 16, 2023
Editorial Decision:	February 21, 2023
Revision Received:	March 9, 2023
Accepted:	March 16, 2023

Editor: Simon Lax

Reviewer(s): Disclosure of reviewer identity is with reference to reviewer comments included in decision letter(s). The following individuals involved in review of your submission have agreed to reveal their identity: Christian Diener (Reviewer #2)

Transaction Report:

DOI: <https://doi.org/10.1128/msystems.00028-23>

February 21, 2023

Mx. Jana C. Massing
Helmholtz-Institut für Funktionelle Marine Biodiversität an der Universität Oldenburg
Oldenburg
Germany

Re: mSystems00028-23 (Quantification of metabolic niche occupancy dynamics in a Baltic Sea bacterial community)

Dear Mx. Jana C. Massing:

Thank you for submitting your revised manuscript to mSystems. It was seen again by the original reviewer as well as an additional reviewer, and I am pleased to inform you that, in principle, we expect to accept it for publication in mSystems. However, acceptance will not be final until you have adequately addressed the reviewers' comments. Please also note that the manuscript is missing ASM's required data availability paragraph (<https://journals.asm.org/open-data-policy>).

Preparing Revision Guidelines

Sincerely,

Simon Lax

Editor, mSystems

Journals Department
Reviewer comments:

Reviewer #1 (Comments for the Author):

In this revised manuscript, Massing et al. address my earlier concerns with the previous version of this manuscript. Specifically, they have added additional text to clarify some of the methodological concerns and they have provided additional definitions. I recognize that this is a sophisticated analysis, and the authors backgrounds suggest a high level of mathematical rigor. I think it is sometimes lost on the authors that the readers of mSystems may not have the same level of mathematical understanding. However, I think the additional text they have provided help resolve some of these issues. I still think that this manuscript will best serve the more statistics savvy mSystems reader and be a struggle for others.

General Comments:

Line 99-137: This is very helpful, and the example is great. Thanks.

Line 174-198: While I appreciate this addition and I think it addresses my original concerns; the new text could be a bit more approachable for a general audience. For example, I find the text on lines 169-173 much easier to understand. The text starting on line 174 seems redundant with some parts of the earlier text. This section could be refined and integrated a bit to increase flow and understanding at the same time.

Line 205: PHATE - I finally just looked this up and learned that it is a visualization approach. Including that information would make this sentence better.

Line 476: I accept the argument that ASV data is the only data available to meet the needs of the method. But perhaps it is worth clearly outlining the data needs (or limitations) of the method. That would allow readers to understand if they can use this in their own work.

Additional Methods (Line 537-605): Thank you.

Reviewer #2 (Comments for the Author):

Summary

In this manuscript, Massing et al. study an amplicon sequencing time series data set from the baltic sea using gene content diffusion map embeddings for the observed taxa. They mapped gene functional groups to the ends of the resulting diffusion paths, interpret those as elementary functional strategies, and investigate how those correspond to environmental measurements.

There is definitely need for approaches that try to infer coarse-scale functional patterns in microbiome data and the strategy used here is creative and does deliver a novel way to do so, along with some limited validation. I think it is a good manuscript with an interesting approach to track functional traits along a well-sampled ecosystem. Some improvements could be made by giving some better intuition why diffusion maps are particularly well suited for genomic data, and by sticking a bit more to statements that are supported by the data at hand.

Major points

I feel that the authors made it difficult on themselves by trying to frame their gene content embedding in terms of niches. The definition of the (ecological) niche is already somewhat contended in the field, and I am not sure whether the embeddings the authors derive for the genomes do indeed represent ecological niches. The abstract defines the niche as "[.] the set of strategies that enable a population to persist and define its impacts on the surroundings", but I would argue that this does not fit any of the contemporary definitions of an ecological niche which all at least maintain some dependence on the environment/habitat and ecological interactions. This is particular true for the Hutchinsonian niche which is cited by the authors (see <https://doi.org/10.1073/pnas.0901650106>). Additionally, even a single strain can change its own (fundamental) niche based on the available resources (for instance oxic vs anoxic growth, availability of different carbon sources, etc.), so the relationship between an organism and its (fundamental) niche is not necessarily a 1:1.

I think it would have been much more straightforward to define the embeddings as "genetic strategies", "evolutionary paths", or "functional traits". Some of those can very well relate to niches, but not all of them, and I think Figure 6A makes this point pretty well (most strategies only correlate weakly with key environmental measures).

If the authors wanted to prove that some of the strategies are indeed driven by niches, they could maybe do so by applying the competitive exclusion principle to their time series and show that even phylogenetically distant taxa can not coexist at a particular timepoint if they share similar strategies.

Also, the title is a bit misleading since no part of the manuscript calculates "niche occupancy" which implies some measure of how many taxa occupy the same niche.

I think the introduction would be more accessible if it would give some intuition why diffusion maps are particularly applicable to gene presence/absence data. Maybe a more intuitive version of the argument made in L121-134 would be that that genomic mutations and rearrangements tend to be small, will usually affect only a few genes, and often lead to inviable states. As such, the real genetic "distance" between one organism and the other is a full trajectory of accumulated small viable genetic modifications, which intuitively fits the basic idea of diffusion maps since a path that crosses many stably existing taxa (viable genetic strategies) is probably more credible than a direct linear path.

Though the paper gives a lot of details on calculating the diffusion maps, it gives very little information about how the gene presence/absence matrices were derived. The description of matching the ASVs to GTDB genomes should be expanded. Which genome did the authors choose when the ASV matched several genomes equally well? Were the genomes summarized to pan-genus models, or was a representative genome chosen? What parameters were used for the BLAST search?

When comparing genes across distant taxa, this usually requires some kind of homology mapping to account for the divergent sequences, but I could not find any details on this process in the text. The methods section states that PROKKA was used for annotation, but this would give only taxon-specific gene names that can't be compared across organisms per se. The repeatedly cited reference [20] reconstructed (gapfilled) metabolic models with CARVME and thus inferred absence/presence maps based on metabolic reactions contained in the BIGG database. Is this what was done here?

Minor points

Figure 1 needs a better caption that explains what is shown in the drawing. What are the circles? Why are some of them larger than others? If genes are quantified only by absence/presence why are the x- and y-axis continuous?

Fig. 4B has legends for "total" and "other" family abundances, but this data is missing from the plot.

L359-361: This statement seems to be incorrect, since there is a higher peak during 2014 in the plot.

Fig. 6A should also show p-values adjusted for multiple testing, either by only showing significant correlations or indicating significant combinations with some symbol. Please provide units for concentrations in the caption.

The data used for panel 6A does not seem to be provided along with the manuscript. This should be added as a supplement or deposited into a public repository.

Fig. 7 is missing an explanation of what the error bars represent.

Manuscript mSystems00028-23 (Quantification of metabolic niche occupancy dynamics in a Baltic Sea bacterial community)

Response to Reviewer Comments

Dear Dr. Lax,

Please see below, in blue, the point-by-point response to the reviewer's comments. All line numbers refer to the revised manuscript file.

Reviewer #1 (Comments for the Author):

In this revised manuscript, Massing et al. address my earlier concerns with the previous version of this manuscript. Specifically, they have added additional text to clarify some of the methodological concerns and they have provided additional definitions. I recognize that this is a sophisticated analysis, and the authors backgrounds suggest a high level of mathematical rigor. I think it is sometimes lost on the authors that the readers of mSystems may not have the same level of mathematical understanding. However, I think the additional text they have provided help resolve some of these issues. I still think that this manuscript will best serve the more statistics savvy mSystems reader and be a struggle for others.

Author response:

We thank the reviewer for the positive assessment and have followed the specific comments below to make the manuscript more accessible.

General Comments:

Line 99-137: This is very helpful, and the example is great. Thanks.

Author response:

We thank the reviewer for their nice words.

Line 174-198: While I appreciate this addition and I think it addresses my original concerns; the new text could be a bit more approachable for a general audience. For example, I find the text on lines 169-173 much easier to understand. The text starting on line 174 seems redundant with some parts of the earlier text. This section could be refined and integrated a bit to increase flow and understanding at the same time.

Author response:

We revised the section and integrated the part, while trying to make it easier to follow and understand.

Line 205: PHATE - I finally just looked this up and learned that it is a visualization approach. Including that information would make this sentence better.

Author response:

Thank you, we added this information.

Line 476: I accept the argument that ASV data is the only data available to meet the needs of the method. But perhaps it is worth clearly outlining the data needs (or limitations) of the method. That would allow readers to understand if they can use this in their own work.

Author response:

We added explanations at the end of the method section “Diffusion mapping the strategy space”.

Additional Methods (Line 537-605): Thank you.

Author response:

We thank the reviewer for their helpful input.

Reviewer #2 (Comments for the Author):

Summary

In this manuscript, Massing et al. study an amplicon sequencing time series data set from the baltic sea using gene content diffusion map embeddings for the observed taxa. They mapped gene functional groups to the ends of the resulting diffusion paths, interpret those as elementary functional strategies, and investigate how those correspond to environmental measurements.

There is definitely need for approaches that try to infer coarse-scale functional patterns in microbiome data and the strategy used here is creative and does deliver a novel way to do so, along with some limited validation. I think it is a good manuscript with an interesting approach to track functional traits along a well-sampled ecosystem. Some improvements could be made by giving some better intuition why diffusion maps are particularly well suited for genomic data, and by sticking a bit more to statements that are supported by the data at hand.

Author response:

We thank the reviewer for the positive assessment and have followed the specific comments below.

Major points

I feel that the authors made it difficult on themselves by trying to frame their gene content embedding in terms of niches. The definition of the (ecological) niche is already somewhat contended in the field, and I am not sure whether the embeddings the authors derive for the genomes do indeed represent ecological niches.

The abstract defines the niche as "[...] the set of strategies that enable a population to persist and define its impacts on the surroundings", but I would argue that this does not fit any of the

contemporary definitions of an ecological niche which all at least maintain some dependence on the environment/habitat and ecological interactions. This is particular true for the Hutchinsonian niche which is cited by the authors (see <https://doi.org/10.1073/pnas.0901650106>).

Author response:

We agree that the term “niche” is used in different contexts and there are different definitions of this term. Pianka et al. 2017 (<https://doi.org/10.1086/693781>) gives this definition of the niche: “The niche, a central concept in ecology, describes not only the environmental conditions required for survival and positive fitness but also the organism’s potential impact on its environment.” and discusses also the multitude of definitions. Kearney et al. 2010 (<https://doi.org/10.1098/rstb.2010.0034>) even argue that functional traits act as the link between environmental factors and organismal fitness.

Certainly, an organism's traits reflect the ways in which the environment impacts their fitness. We would argue that our definition of the niche fits well with contemporary definitions and approaches of using combinations of traits and strategies to define or classify the niche space (e.g. Winnemiller et a. 2015, Carmona et al. 2016, Koutsidi et al. 2020). In general, we think that “niche” is still a useful term, especially to strengthen understanding of the links of individual species and their traits and capabilities to ecological roles and ecosystem functions.

Traits and strategies that the species deploy to survive and reproduce are closely linked to their environment/habitat and can define interactions between species. A certain substrate preference for example defines the requirements of the species as well as the impact and part of its ecological role in the ecosystem. Adaptations to living associated with a host define this interaction with the other species.

Additionally, even a single strain can change its own (fundamental) niche based on the available resources (for instance oxic vs anoxic growth, availability of different carbon sources, etc.), so the relationship between an organism and its (fundamental) niche is not necessarily a 1:1.

Author response:

We agree that deployed strategies can change depending on a given set of conditions. Since our input data is genomes, we only analyze the set of theoretical capabilities, but we envision that with future transcriptomic data it could also be possible to investigate the strategies that are deployed under a given set of environmental conditions. We added this information to the discussion.

I think it would have been much more straightforward to define the embeddings as "genetic strategies", "evolutionary paths", or "functional traits". Some of those can very well relate to niches, but not all of them, and I think Figure 6A makes this point pretty well (most strategies only correlate weakly with key environmental measures).

Author response:

The diffusion map finds new variables that are composites of metabolic capabilities of the analyzed taxa. We would argue that the term “metabolic strategy” describes these new variables well.

Figure 6A illustrates that seasonality, hence the change of environmental conditions is indeed correlated to the strategy time-series.

If the authors wanted to prove that some of the strategies are indeed driven by niches, they could maybe do so by applying the competitive exclusion principle to their time series and show that even phylogenetically distant taxa can not coexist at a particular timepoint if they share similar strategies.

Also, the title is a bit misleading since no part of the manuscript calculates "niche occupancy" which implies some measure of how many taxa occupy the same niche.

Author response:

We show that the coarse-grained metabolic strategies are indeed shared by different taxa and different taxa can dominate the strategy over the year. Hence, also the title describes well that we show the metabolic strategies/ niches and how they are occupied, which taxa are dominating or exhibiting this strategy.

I think the introduction would be more accessible if it would give some intuition why diffusion maps are particularly applicable to gene presence/absence data. Maybe a more intuitive version of the argument made in L121-134 would be that that genomic mutations and rearrangements tend to be small, will usually affect only a few genes, and often lead to inviable states. As such, the real genetic "distance" between one organism and the other is a full trajectory of accumulated small viable genetic modifications, which intuitively fits the basic idea of diffusion maps since a path that crosses many stably existing taxa (viable genetic strategies) is probably more credible than a direct linear path.

Author response:

This is an interesting point. However, we would argue that we investigate the differences in functional capabilities (known gene content), which is not identical with genetic distance. The main argument for the use of the diffusion map is that long-distance comparisons are hard to make because we do not know the variables that we should use to compare very different objects (L107-120). The metabolic strategies that we want to detect are often nonlinear combinations of the metabolic capabilities of the taxa.

Though the paper gives a lot of details on calculating the diffusion maps, it gives very little information about how the gene presence/absence matrices were derived. The description of matching the ASVs to GTDB genomes should be expanded. Which genome did the authors choose when the ASV matched several genomes equally well? Were the genomes summarized to pan-genus models, or was a representative genome chosen? What parameters were used for the BLAST search?

Author response:

We added explanations in the method section "Obtaining genomes and genes from ASV data".

When comparing genes across distant taxa, this usually requires some kind of homology mapping to account for the divergent sequences, but I could not find any details on this process

in the text. The methods section states that PROKKA was used for annotation, but this would give only taxon-specific gene names that can't be compared across organisms per se. The repeatedly cited reference [20] reconstructed (gapfilled) metabolic models with CARVME and thus inferred absence/presence maps based on metabolic reactions contained in the BIGG database. Is this what was done here?

Author response:

We annotated the obtained genomes using Prokka (see section 'Obtaining genomes and genes from ASV data'). From these annotations we derived our absence/presence matrix of all annotated genes in all genomes. We are not interested in the evolutionary relationships of the taxa, instead our aim here is to compare their metabolic capabilities.

Minor points

Figure 1 needs a better caption that explains what is shown in the drawing. What are the circles? Why are some of them larger than others? If genes are quantified only by absence/presence why are the x- and y-axis continuous?

Author response:

We apologize that we did not explain this in detail. The circles represent the genomes, which have different abundances over time (size). The two axes in the figure are just representatives for the many axes that describe the known gene content of the genomes in a high-dimensional space. We changed the axes names accordingly and provided explanations in the caption.

Fig. 4B has legends for "total" and "other" family abundances, but this data is missing from the plot.

Author response:

The data is represented as the black line and the grey dotted line. Maybe they were not visible as the quality of the figure was too low for the first submission. We apologize and will provide a high-quality image for this submission.

L359-361: This statement seems to be incorrect, since there is a higher peak during 2014 in the plot.

Author response:

Thank you, we corrected the respective sentence.

Fig. 6A should also show p-values adjusted for multiple testing, either by only showing significant correlations or indicating significant combinations with some symbol. Please provide units for concentrations in the caption.

Author response:

We modified Fig. 6A to include the adjusted p-values and provided the concentration units in the caption.

The data used for panel 6A does not seem to be provided along with the manuscript. This should be added as a supplement or deposited into a public repository.

Author response:

The data set is from a related project and can be obtained from authors upon request. We have added this information.

Fig. 7 is missing an explanation of what the error bars represent.

Author response:

The error bars represent the standard error of the functional diversity estimation as indicated in the caption.

March 16, 2023

Mx. Jana C. Massing
Helmholtz-Institut für Funktionelle Marine Biodiversität an der Universität Oldenburg
Oldenburg
Germany

Re: mSystems00028-23R1 (Quantification of metabolic niche occupancy dynamics in a Baltic Sea bacterial community)

Dear Mx. Jana C. Massing:

Thank you for your thoughtful responses to the two reviewers' assessments. I am pleased to say that your manuscript has been accepted, and I am forwarding it to the ASM Journals Department for publication. For your reference, ASM Journals' address is given below. Before it can be scheduled for publication, your manuscript will be checked by the mSystems production staff to make sure that all elements meet the technical requirements for publication. They will contact you if anything needs to be revised before copyediting and production can begin. Otherwise, you will be notified when your proofs are ready to be viewed.

If you would like to submit a potential Featured Image, please email a file and a short legend to mSystems@asmusa.org. Please note that we can only consider images that (i) the authors created or own and (ii) have not been previously published. By submitting, you agree that the image can be used under the same terms as the published article. File requirements: square dimensions (4" x 4"), 300 dpi resolution, RGB colorspace, TIF file format.

We recognize that the video files can become quite large, and so to avoid quality loss ASM suggests sending the video file via <https://www.wetransfer.com/>. When you have a final version of the video and the still ready to share, please send it to mSystems staff at mSystems@asmusa.org.

Sincerely,

Simon Lax
Editor, mSystems

Journals Department
E-mail: mSystems@asmusa.org